# Type II and IV toxin-antitoxin systems coordinately stabilize the integrative and conjugative element of the ICE*Sa*2603 family conferring multiple drug resistance in *Streptococcus suis*

**Qibing Gu**[1,2,3], **Xiayu Zhu**[1,2,3], **Yong Yu**[1,2,3], **Tao Jiang**[4], **Zihao Pan**[1,2,3]*, **Jiale Ma**[1,2,3]*, **Huochun Yao**[1,2,3]*

**1** MOE Joint International Research Laboratory of Animal Health and Food Safety, College of Veterinary Medicine, Nanjing Agricultural University, Nanjing, China, **2** Key Lab of Animal Bacteriology, Ministry of Agriculture, Nanjing, China, **3** OIE Reference Lab for Swine Streptococcosis, Nanjing, China, **4** Department of Stomatology, Jinling Hospital, Affiliated Hospital of Medical School, Nanjing University, Nanjing, China

\* panzihao@njau.edu.cn (ZP); jialema@njau.edu.cn (JM); yaohch@njau.edu.cn (HY)

**Data Availability Statement:** All data are in the manuscript and Supporting information files.

## Abstract

Integrative and conjugative elements (ICEs) play a vital role in bacterial evolution by carrying essential genes that confer adaptive functions to the host. Despite their importance, the mechanism underlying the stable inheritance of ICEs, which is necessary for the acquisition of new traits in bacteria, remains poorly understood. Here, we identified SezAT, a type II toxin-antitoxin (TA) system, and AbiE, a type IV TA system encoded within the ICE*S-su*HN105, coordinately promote ICE stabilization and mediate multidrug resistance in *Streptococcus suis*. Deletion of SezAT or AbiE did not affect the strain's antibiotic susceptibility, but their duple deletion increased susceptibility, mainly mediated by the antitoxins SezA and AbiEi. Further studies have revealed that SezA and AbiEi affect the genetic stability of ICE*S-su*HN105 by moderating the excision and extrachromosomal copy number, consequently affecting the antibiotic resistance conferred by ICE. The DNA-binding proteins AbiEi and SezA, which bind palindromic sequences in the promoter, coordinately modulate ICE excision and extracellular copy number by binding to sequences in the origin-of-transfer (*oriT*) and the *attL* sites, respectively. Furthermore, AbiEi negatively regulates the transcription of SezAT by binding directly to its promoter, optimizing the coordinate network of SezAT and AbiE in maintaining ICE*Ssu*HN105 stability. Importantly, SezAT and AbiE are widespread and conserved in ICEs harbouring diverse drug-resistance genes, and their coordinated effects in promoting ICE stability and mediating drug resistance may be broadly applicable to other ICEs. Altogether, our study uncovers the TA system's role in maintaining the genetic stability of ICE and offers potential targets for overcoming the dissemination and evolution of drug resistance.

**Funding:** This work was supported by the National Natural Science Foundation of China (HY, No. 31972650), and the Postgraduate Research & Practice Innovation Program of Jiangsu Province (QG, No. KYCX22_0780). The funders had no role in study design, data collection and analysis, decision to publish, or preparation of the manuscript.

**Competing interests:** The authors have declared that no competing interests exist.

## Author summary

Integrative and conjugative elements (ICEs) are mobile genetic elements that play a crucial role in driving bacterial evolution and the acquisition of new traits, including multidrug resistance. An essential prerequisite for bacteria to acquire new features is the stable inheritance of ICEs. In this study, we demonstrated that the type II TA system SezAT and the type IV TA system AbiE play a coordinated role in the genetic stabilization of ICE*Ssu*HN105. SezAT and AbiE coordinately promote multidrug resistance in *S. suis*, an effect mediated by the antitoxins. Our findings suggest that this cooperative mechanism may have broader implications for other ICEs. These revelations contribute to our comprehension of the physiological role of the TA systems and the genetic stabilization mechanisms of ICEs.

## Introduction

The toxin-antitoxin (TA) system, crucial for bacterial survival under stress, comprises a stable toxin and an unstable antitoxin [1]. Based on the nature and mode of action of the antitoxin, TA systems are now classified into eight categories [2]. The type II TA system is the most studied: both the antitoxin and the toxin are proteins, and the antitoxin neutralizes the toxicity of the toxin through direct interaction. In the type IV TA system, both toxins and antitoxins are proteins. The antitoxin deactivates the toxin by competing with it for the target. The antitoxin is generally a DNA-binding protein that binds to the palindromic sequence in the promoter of the TA system to achieve auto-regulation, which contributes to the rapid release of the toxin under stressful conditions to perform its physiological function [3]. By binding to the promoter of genes, antitoxins also have the ability to control the transcription of other genes [4]. The TA systems were originally discovered on conjugative plasmids and were later thought to maintain plasmid stability through post-segregational killing (PSK) [5,6]. PSK ensures plasmid stability by killing cells that lose their plasmids after division [7]. However, this hypothesis remains controversial and does not fully explain the stability of the plasmid encoding multiple TA systems [8]. Furthermore, it is proposed that MosAT contributes to the maintenance of ICE SXT (ICE carrying sulfamethoxazole and trimethoprim resistance) stabilization [9], but the exact mechanism is unclear. Further exploration of the role of the TA system in maintaining the stability of mobile genetic elements will not only expand the physiological functions of the TA system but also help us comprehend the mechanisms underlying the evolution of bacterial drug resistance.

Integrative and conjugative elements (ICEs) are mobile genetic elements that can transfer widely among bacteria and play a significant role in bacterial evolution. ICEs encode genes essential for both integrating into and excising from the bacterial genome [10]. The modular nature of ICEs has significant implications for the evolution of bacterial diversity and adaptation. ICEs facilitate the horizontal transfer of a multitude of genes between bacteria via conjugation machinery, thereby conferring advantageous traits such as antibiotic resistance, metal resistance, and virulence upon the recipient host [11,12]. The escalating issue of drug resistance, worsened by the existence of ICE, poses a significant threat to human life and health globally. The horizontal transfer of ICE provides a vehicle and impetus for the prevalence of resistance, and the horizontal transfer mechanisms of various ICEs have garnered extensive attention for many years. SetR encoded in SXT was shown to control the transfer of SXT by binding to four operators [13]. The activation of the ICE*clc* of *Pseudomonas putida* was observed in stationary phase cells, with excision and transfer primarily occurring following the

introduction of fresh nutrients [14]. However, the mechanism underlying the stable inheritance of ICE in bacteria, which is an essential prerequisite for the acquisition of resistance, remains poorly understood.

*Streptococcus suis* (*S. suis*) is an important zoonotic pathogen responsible for human infections such as meningitis, septicemia, arthritis, and endocarditis [15]. Given its widespread presence in the environment, alongside extensive antibiotic use in both clinical and breeding practices for primarily treating, it could serve as an important reservoir of resistance, driving the spread and evolution of antibiotic resistance [16]. The abundance of ICESa2603 family homologues in *S. suis* promotes drug resistance [17,18], and in particular, the *erm*(B) and *tet* (O) resistance determinants carried by ICE lead to high levels of resistance to erythromycin, lincomycin, and tetracycline in *S. suis* [19]. The ICESa2603 family of ICEs is widely distributed among pathogenic *Streptococcus* species and plays a significant role in the acquisition of drug resistance [16]. They can be stably integrated into the genome, but the mechanism for maintaining stability remains unclear. A better understanding of these mechanisms could help prevent and control multidrug resistance in streptococci.

In the present study, we revealed that the type II TA system SezAT and the type IV TA system AbiE, encoded by the ICESa2603 family of ICESsuHN105, coordinately promote the genetic stability of ICE and mediate multiple drug resistance. Single deletion of SezAT or AbiE did not affect antibiotic resistance, but the strain's susceptibility to antibiotics significantly increased upon the double deletion of SezAT and AbiE, which was mediated by the antitoxins SezA and AbiEi. AbiEi and SezA bind directly to sequences in the origin of transfer (*oriT*) site and the *attL* site, respectively, to control the genetic stability of ICE and confer resistance. The coordination mechanism of type II and type IV TA systems enhances our comprehension of the intricate physiological functions of TA systems and the evolutionary mechanisms of bacterial resistance.

## Results

### Identifying two TA systems, SezAT and AbiE, encoded in ICE*Ssu*HN105 harbouring diverse drug resistance genes

Multiple TA systems are present in the genome of *S. suis*, however, their biological roles remain incompletely elucidated. To investigate their function, we conducted an alignment analysis of the proteins encoded by the genome of *S. suis* serotype 5 strain HN105. The protein DF184_05140 was determined to be a homolog of the type II TA toxin SezT encoded in *S. suis* strain 05ZYH33 with an 89.53% sequence identity, and its downstream DF184_05135 displayed a 92.26% sequence identity with the antitoxin SezA in strain 05ZYH33 [20]. Therefore, we adopted the nomenclature in 05ZYH33 and designated the proteins DF184_05135-DF184_05140 as SezAT. Unexpectedly, the proteins DF184_04900-DF184_04905 encoded in the genetic neighborhood share sequence identities of 92% and 88.27% with the toxin AbiEii and antitoxin AbiEi, which constitute the type IV TA system AbiE in *Streptococcus agalactiae* [21]. Furthermore, DF184_04865 and DF184_05095, encoded by two non-contiguous genes, were identified as homologues of toxin zeta and antitoxin epsilon, respectively, which typically constitute a TA system in other bacteria [22]. These three TA systems are situated within the ICE*Ssu*HN105 [18], which harbors a variety of resistance genes (Fig 1A). Next, we analyzed the toxic effects of the three aforementioned toxins and the detoxification of their antitoxins using an arabinose-induced *E. coli* expression system by enumerating the colony-forming units (CFUs) through dilution-plating and measuring the $OD_{600}$ values. We found that the production of the toxin SezT significantly induced bacterial death, whereas co-expression with its antitoxin SezA neutralized the toxicity of SezT (Fig 1B

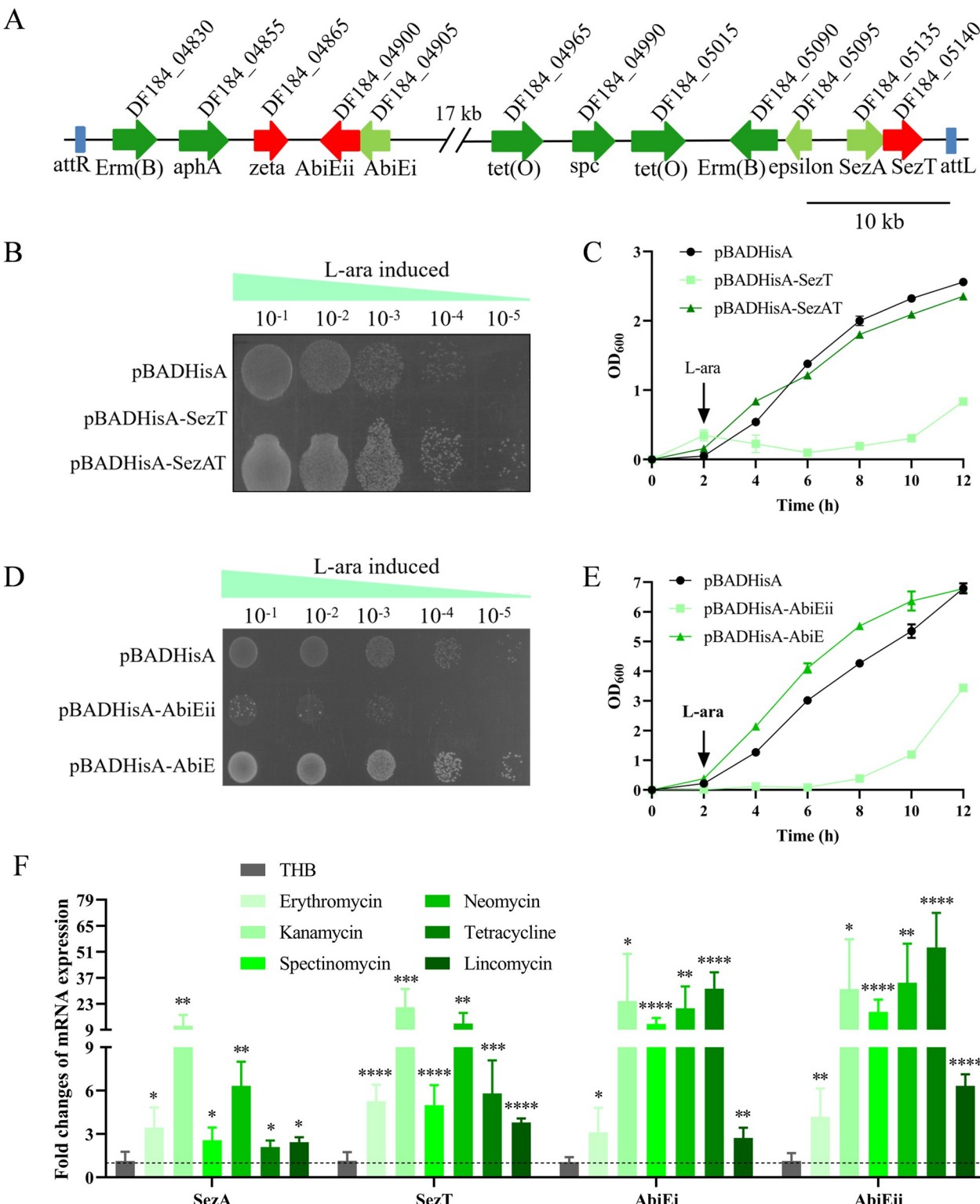

**Fig 1. Identification of the Type II TA system, SezAT, and the Type IV TA system, AbiE, in ICES*su*HN105.** (A) Schematic representation of the ICES*su*HN105. Three TA systems, SezAT, AbiE and zeta-epsilon, and several resistance genes, Erm(B), aphA, tet(O) and spc, are encoded in ICES*su*HN105. (B) CFUs were analyzed for the toxicity of SezT and the ability of SezA to neutralize toxicity. (C) Growth curves for cells transfected with pBADHisA, pBADHisA-SezT or pBADHisA-SezAT vectors were determined under L-arabinose induction. (D) CFUs were analyzed for the toxicity of AbiEii and the ability of AbiEi to neutralize toxicity. (E) Growth curves for cells transfected with pBADHisA, pBADHisA-AbiEii or

pBADHisA-AbiE vectors were determined under L-arabinose induction. (F) Activation of SezAT and AbiE under various antibiotics. All experiments were conducted independently three times. Unpaired two-tailed Student's t-test: * $P < 0.05$; ** $P < 0.01$; *** $P < 0.001$; **** $P < 0.0001$.

and 1C). These results suggest that SezAT, encoded on ICE*Ssu*HN105, is a functional type II TA system. Similarly, AbiE was also identified as a functional TA system (Fig 1D and 1E). However, the zeta-epsilon is inactive (S1A and S1B Fig). Functional domain analysis of AbiEi revealed a lack of ribonuclease activity and post-translational modification function [7], thereby excluding AbiE as a type V or type VII TA. The bacterial two-hybrid experiment revealed no interaction between AbiEi and AbiEii, effectively ruling out the possibility that AbiE is a type II TA or type VI TA (S1G Fig) [7]. The structures of the antitoxin AbiEi and the toxin AbiEii were predicted using AlphaFold, which showed that AbiEi overlaps almost completely with the antitoxin AbiEi (6y8q.1.A) of the type IV TA system in *S. agalactiae* (S1C Fig). The structure of AbiEii partially overlaps with the toxin TglT (6j7q.1.A) in *Mycobacterium tuberculosis* (S1D Fig). Taken together, AbiE encoded on ICE*Ssu*HN105 is a type IV TA system. We further characterized the AbiE system using the co-transformation approach to evaluate the usability of the pBAD33 and pET28a co-transformation method. The results showed that toxin AbiEii induced cell death in the pBAD33-AbiEii and pET28a co-transformed group after the addition of the inducers, while there was no significant difference in the CFUs in the pBAD33-AbiEii and pET28a-AbiEi co-transformed group compared to the control group (pBAD33/pET28a), suggesting that this method is effective in identifying the toxic effects of toxin AbiEii and the detoxification of antitoxin AbiEi, and can be used to identify the TA system (S1E and S1F Fig).

## SezAT and AbiE coordinately facilitate the multiple drug resistance in *S. suis* strain HN105

Numerous studies suggest that TA systems contribute to bacteria against various stresses, such as antibiotic treatment [23–26]. SezAT and AbiE are located in the ICE*Ssu*HN105, which harbors diverse resistance genes that mediate the acquisition of multidrug resistance in HN105 [18]. We subsequently treated HN105 with six different antibiotics, and the transcript levels of SezAT and AbiE were determined after 6 hours of treatment. The findings indicated a significant upregulation in the transcript levels of both SezAT and AbiE post-antibiotic treatment in comparison to pre-treatment levels, implying a potentially pivotal role of SezAT and AbiE in antibiotic resistance (Fig 1F). Subsequently, deletion strains Δ*SezAT* and Δ*AbiE* were constructed and the minimum inhibitory concentrations (MICs) of multiple drugs were assessed. The single deletion of SezAT or AbiE did not alter the strain's susceptibility to antibiotics. However, the MICs of antibiotics were significantly reduced following the double deletion of SezAT and AbiE, suggesting that SezAT and AbiE coordinately facilitate the acquisition of multiple drug resistance (Table 1). To explore the function of toxins and antitoxins in the multidrug resistance of the HN105 strain, we next constructed single deletion strains of toxins and antitoxins. The results showed that neither the deletion of toxin nor antitoxin changed the MICs of the antibiotics. Next, we constructed the double deletion strains Δ*SezT-AbiEii* and Δ*SezA-AbiEi*. The MICs of antibiotics against Δ*SezT-AbiEii* and Δ*SezA-AbiEi* showed that double deletion of the antitoxin resulted in a significant increase in the susceptibility of the strain to the antibiotics, whereas double deletion of the toxin had no such effect (Table 1). These data indicate that the promotion of multidrug resistance in HN105 by SezAT and AbiE is coordinately mediated by the antitoxins SezA and AbiEi.

**Table 1. The MICs of strain to antibiotics.**

| Antibiotics | Kanamycin | Neomycin | Erythromycin | Tetracycline | Spectinomycin | Lincomycin |
|---|---|---|---|---|---|---|
| | MIC (u g/ml) | | | | | |
| HN105 | 8192 | 1024 | 4096 | 64 | 16384 | 1024 |
| ΔSezAT | 8192 | 1024 | 4096 | 64 | 16384 | 1024 |
| ΔAbiE | 8192 | 1024 | 4096 | 64 | 16384 | 1024 |
| **ΔSezAT-AbiE** | **128** | **128** | **<32** | **<4** | **64** | **<32** |
| ΔSezA | 8192 | 1024 | 4096 | 64 | 16384 | 1024 |
| ΔAbiEi | 8192 | 1024 | 4096 | 64 | 16384 | 1024 |
| **ΔSezA-AbiEi** | **64** | **128** | **<32** | **<4** | **64** | **<32** |
| ΔSezT | 8192 | 1024 | 4096 | 64 | 16384 | 1024 |
| ΔAbiEii | 8192 | 1024 | 4096 | 64 | 16384 | 1024 |
| ΔSezT-AbiEii | 8192 | 1024 | 4096 | 64 | 16384 | 1024 |

## Either SezA or AbiEi is required for the genetic stability of ICE*su*HN105

The increase of antibiotic susceptibility upon TA system double deletion and antitoxin double deletion suggested that SezAT and AbiE may mediate multidrug resistance by controlling the genetic stability of ICE*su*HN105. ICE*su*HN105 is capable of being excised from the chromosome, subsequently forming a circular extrachromosomal DNA molecule [18]. To examine the impact of SezAT and AbiE on the integration and excision of ICE*su*HN105, four specific primers (P1, P2, P3, and P4) were devised and utilized in various combinations to identify the integrated (P1/P2 and P3/P4 positive; P2/P3 and P1/P4 negative) and excised (P1/P2 and P3/P4 negative; P2/P3 and P1/P4 positive) states of ICE*su*HN105 in HN105 (Fig 2A). PCR analysis was initially conducted on mutant strains Δ*SezAT*, Δ*AbiE*, and Δ*SezAT-AbiE*. Amplification of products was discernible in the wild-type HN105, Δ*SezAT*, and Δ*AbiE* strains, while only the P1/P4 amplified product was detected in the Δ*SezAT-AbiE* strain, with the remaining three pairs of primers showing no amplification (Fig 2B). The results showed that only ICE*su*HN105 excision from the chromosome could be detected in the Δ*SezAT-AbiE* group, while the integrated and circular forms of ICE*su*HN105 could not be detected, indicating that the double deletion of SezAT and AbiE resulted in the loss of ICE*su*HN105. Furthermore, the primer design was employed to amplify attI/attB/hydrodase, and three standard curves were generated for quantifying the copy number of attI/attB/hydrodase through qPCR (S2 Fig). The excision frequency and extrachromosomal copy number of ICE*su*HN105 were subsequently assessed in accordance with the methodology outlined in reference [27]. The result showed that the excision frequency (>100%) of ICE*su*HN105 in Δ*SezAT-AbiE* was greatly increased compared to the wild strain (Fig 2C). Theoretically, one attB corresponds to one copy number of hydrodase, so the excision frequency should be 100% when ICE is in a fully excised, non-integrated state. Our findings demonstrate that the ICE*su*HN105 was completely excised after double deletion of SezAT and AbiE. Meanwhile, the extrachromosomal copy number of ICE*su*HN105 in the Δ*SezAT-AbiE* strain was significantly reduced to nearly zero compared to HN105 (Fig 2D). The combined data suggested that SezAT and AbiE coordinately promote the genetic stability of ICE*su*HN105.

Δ*SezA*, Δ*AbiEi*, Δ*SezA-AbiEi*, Δ*SezT*, Δ*AbiEii*, and Δ*SezT-AbiEii* were constructed to determine the role of toxins and antitoxins in the stable inheritance of ICE*su*HN105. PCR analysis showed that only ICE excision from the chromosome was detected following double deletion of the antitoxins SezA and AbiEi, and the integrated and circular forms of ICE*su*HN105 were undetectable (Fig 2E). However, the double deletion of toxins did not affect the integration,

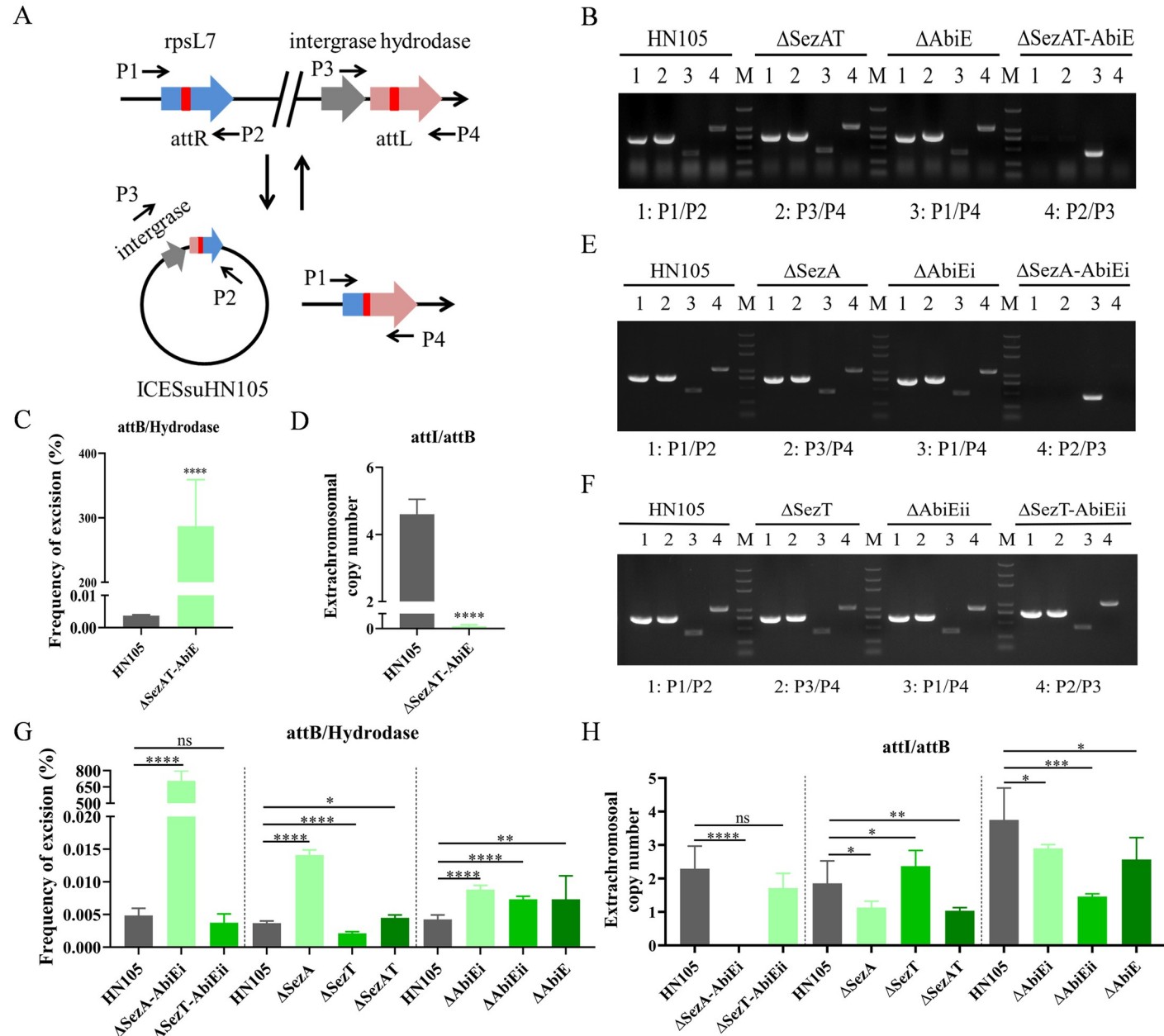

**Fig 2. SezAT and AbiE synergistically maintain the genetic stability of ICE*Ssu*HN105 through antitoxins.** (A) Schematic representation of the excision and integration of ICE*Ssu*HN05. P1, P2, P3 and P4 represent the positions of the excision and circular form detection primers, respectively. *attL* and *attR* indicate the integration sites of ICE*Ssu*HN105 in the chromosome. (B) PCR analysis of excision and circular forms of ICE*Ssu*HN105 in HN105, Δ*SezAT*, Δ*AbiE* and Δ*SezAT-AbiE*. (C) The excision frequency of ICE*Ssu*HN105 in Δ*SezAT-AbiE* was determined by qPCR. (D) Extrachromosomal copy number of ICE*Ssu*HN105 in Δ*SezAT-AbiE* was determined by qPCR. (E) PCR analysis of excision and circular forms of ICE*Ssu*HN105 in HN105, Δ*SezA*, Δ*AbiEi* and Δ*SezA-AbiEi*. (F) PCR analysis of excision and circular forms of ICE*Ssu*HN105 in HN105, Δ*SezT*, Δ*AbiEii* and Δ*SezT-AbiEii*. (G) Analysis of the excision frequency of ICE*Ssu*HN105 in deletion strains. (H) Analysis of extrachromosomal copy number of ICE*Ssu*HN105 in deletion strains. All experiments were conducted independently three times. Unpaired two-tailed Student's t-test: ns p > 0.05; * P < 0.05; ** P < 0.01; *** P < 0.001; **** P < 0.0001.

excision, and circular molecule formation of ICE*Ssu*HN105 (Fig 2F). These results, in conjunction with the antibiotic-resistant phenotypes presented in Table 1, demonstrate that the genetic stability of ICE*Ssu*HN105 is mediated by the antitoxins, not the toxins. The excision frequency and extrachromosomal copy number of ICE in deletion strains were further assessed.

Consistent with the previous PCR results, ICE*Ssu*HN105 was completely excised and lost when SezA and AbiEi were double deleted, whereas toxins double deletion did not affect the excision frequency and extrachromosomal copy number of ICE*Ssu*HN105 (Fig 2G and 2H). Deletion of either the antitoxin SezA or AbiEi resulted in a notable increase in excision frequency and a decrease in extrachromosomal copy number, consistent with deletions in the TA system (Fig 2G and 2H). Conversely, the deletion of SezT led to a substantial decrease in excision frequency and a significant increase in extrachromosomal copy number, which was attributed to the up-regulation of the transcription level of SezA following SezT deletion (S3A Fig). Excision frequency was significantly increased after AbiEii deletion, and the diminished extrachromosomal copy number was attributed to a significant decrease in the transcription level of the antitoxin AbiEi (S3B Fig). These data suggested that both the antitoxins SezA and AbiEi are required for the genetic stability of ICE*Ssu*HN105.

### Identifying AbiEi as a regulator blocking the *oriT* site to maintain the genetic stability of ICESsuHN105

TA systems are typically auto-regulatory, achieved by antitoxin binding to palindromic sequences in the promoter [8]. The antitoxin AbiEi was predicted to be a regulatory factor, with two palindromic sequences identified in the AbiE promoter, designated as IR1 and IR2 (Fig 3A). The confirmation of AbiEi binding to the promoter was achieved through electrophoretic mobility shift assays (EMSA), while analysis of β-galactosidase activity indicated that AbiEi negatively regulates the transcription of AbiE (Fig 3B and 3F). To test whether AbiEi binds to IR1/2, IR1 and IR2 were deleted, followed by EMSA and β-galactosidase activity analyses. AbiEi was able to bind to the promoter when either IR1 or IR2 was absent, but not when both were deleted, indicating that AbiEi binds to IR1 and IR2 in the promoter and it can bind to either IR1 or IR2 alone (Fig 3C–3E). However, the deletion of neither IR1 nor IR2 enabled AbiEi to achieve negative regulation, indicating that the integrity of the IR sequence is required for AbiEi to exert negative regulation (Fig 3F). Collectively, these results demonstrated that AbiEi binds to IR1 and IR2 in the promoter and plays a negative regulatory role, consistent with AbiEi in *S. agalactiae* [21].

The integration and excision of ICE require the involvement of key proteins such as integrase and relaxase [10]. To examine whether AbiE maintains the genetic stability of ICE*Ssu*HN105 by regulating the transcription of these key genes through antitoxin, we analyzed the transcript levels of the genes in Δ*AbiE*. The deletion of AbiE did not affect the transcript levels of genes encoding integrases and relaxase, among others, suggesting that the antitoxin AbiEi may not maintain the genetic stability of ICE by regulating the transcription of target genes (S4 Fig). Given that the antitoxin of type IV TA can compete for binding with the toxin's target [7], we speculated that AbiEi might interact with proteins such as integrases to affect the genetic stability of ICE. However, bacterial two-hybrid analysis confirmed that AbiEi did not interact with proteins associated with ICE integration and excision. *OriT* plays a significant role in the life cycle of ICEs as it is recognized by the ICE-encoded proteins [10]. In addition to relaxase, other DNA-binding proteins can also bind to *oriT* [28]. Analysis of *oriT* and vicinity DNA sequences revealed sequence with high similarity to the palindrome sequences IR1/2 of the AbiE promoter (Fig 3G). Subsequently, the T sequence encompassing *oriT* was amplified and subjected to EMSA to validate the binding capability of AbiEi to *oriT* (S5A Fig). AbiEi bound specifically to the T sequence and the binding capacity increased in a dose-dependent fashion (Fig 3H). Upon deletion of the *oriT* site, AbiEi was unable to bind to the T sequence, confirming its specificity for the *oriT* site (Fig 3I). The binding affinity of AbiEi for the T sequence was analyzed by determining the equilibrium dissociation constant ($K_D$), which was

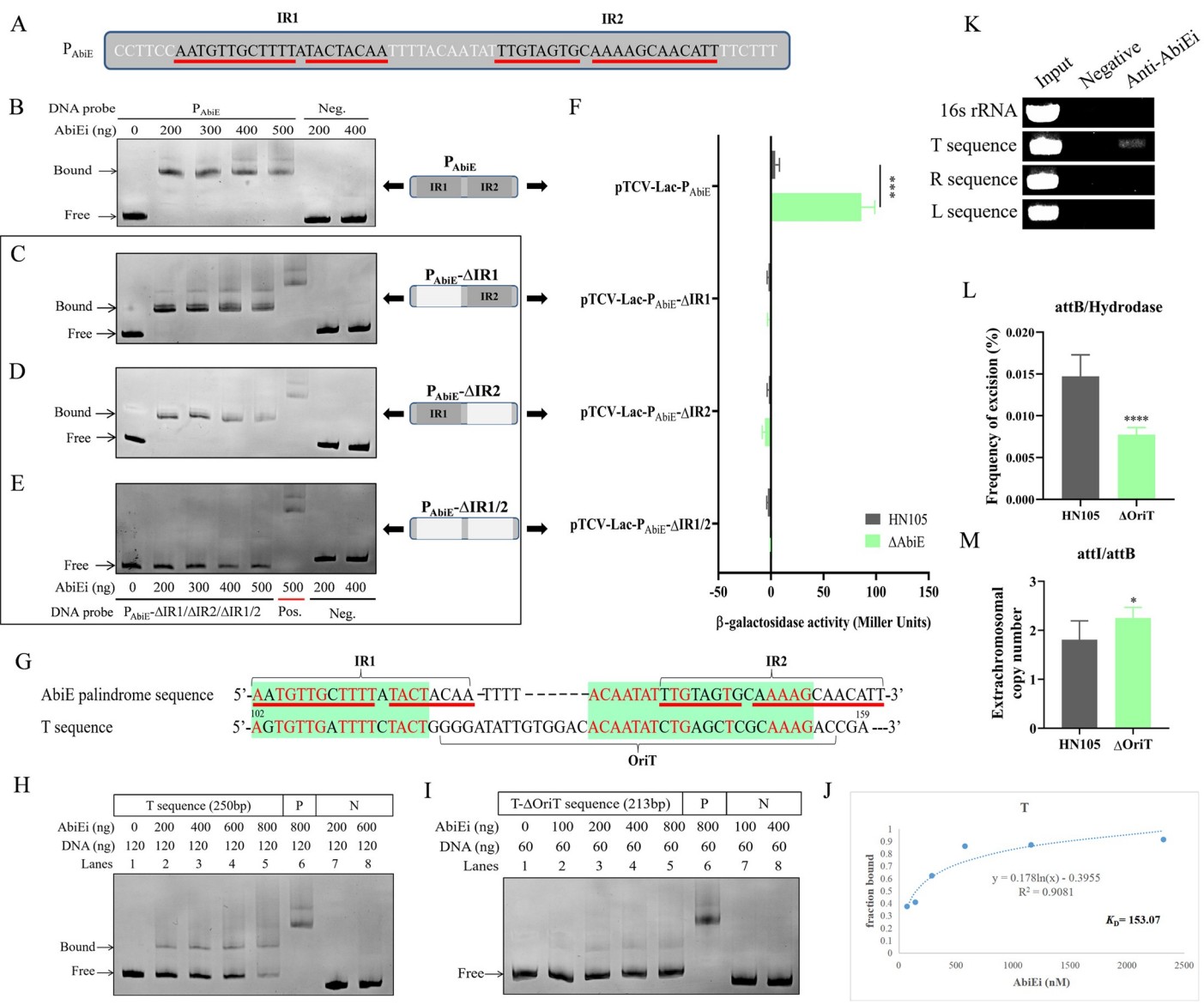

**Fig 3. Auto-regulatory factor AbiEi maintains the genetic stability of ICE*Ssu*HN105 by binding to sequences in the *oriT* site.** (A) Schematic representation of the AbiE promoter palindromic sequences IR1 and IR2. The red underlines indicate the position of the palindromic sequences. EMSA results shown the binding of AbiEi to the AbiE promoter (B), $P_{AbiE}$-ΔIR1 (C), $P_{AbiE}$-ΔIR2 (D) and no binding to $P_{AbiE}$-ΔIR1/2 (E). Different concentrations of AbiEi protein were added to each reaction mixture. DNA probes containing the fragments were used at 120 ng per reaction mixture. Fragments amplified from 16S rRNA were used as negative control. The AbiE promoter fragment served as positive control. (F) pTCV-*lac*-$P_{AbiE}$, pTCV-*lac*-$P_{AbiE}$-ΔIR1, pTCV-*lac*-$P_{AbiE}$-ΔIR2, and pTCV-*lac*-$P_{AbiE}$-ΔIR1/2 were integrated to HN105 and Δ*AbiE* host to determine promoter activity, respectively. (G) Sequence similarity analysis of AbiE promoter IR1 and IR2 with the T sequence at *oriT*. The red underlines indicate the position of the palindromic sequences. (H) EMSA result shown that AbiEi binds to the T sequence. The concentrations of AbiEi were 0 (0 ng), 289.86 (200 ng), 579.71 (400 ng), 869.57 (600 ng), 1159.4 (800 ng), 1159.4 (800 ng), 289.86 (200 ng) and 869.57 nM (600 ng). The AbiE promoter fragment served as positive control. (I) EMSA result shown that AbiE does not bind to the T-ΔoriT sequence. DNA probes containing T-ΔoriT were used at 60 ng per reaction mixture. The concentrations of AbiEi were 0 (0 ng), 144.93 (100 ng), 289.86 (200 ng), 579.71 (400 ng), 1159.4 (800 ng), 1159.4 (800 ng), 144.93 (100 ng) and 579.71 nM (400 ng). The T sequence served as positive control. (J) Binding affinity analysis of AbiEi protein for T sequence. The equilibrium dissociation constant ($K_D$) was determined via EMSA. The AbiEi concentrations used for EMSA were 0 (0 ng), 72.46 (50 ng), 144.93 (100 ng), 289.86 (200 ng), 579.71 (400 ng), 1159.4 (800 ng) and 2318.8 nM (1600 ng). (K) ChIP analysis was performed to detect the binding of AbiE to T, R, and L sequences. Normal mouse serum was used as a negative control. The 16S rRNA gene PCR product was used as a negative control. (L) The excision frequency of ICE*Ssu*HN105 in Δ*oriT* was determined by qPCR. (M) Analysis of extrachromosomal copy number of ICE*Ssu*HN105 in Δ*oriT*. All experiments were conducted independently three times. Unpaired two-tailed Student's t-test: * P < 0.05; *** P < 0.001; **** P < 0.0001.

valued at 153.07 nM (Fig 3J). Chromatin immunoprecipitation (ChIP) analysis further confirmed the binding of AbiEi to the T sequence, and demonstrated that AbiEi could also directly bind to the T sequence *in vivo* (Fig 3K). The *attL* and *attR* sites are two specific loci where the ICE is integrated into the chromosome and are recognized by the recombinase [10]. To investigate the potential binding of AbiEi to *attL/R* sites, sequences containing these sites were designated as L and R sequences, respectively (S5B and S5C Fig). However, ChIP analysis showed no binding of AbiEi to these sequences, indicating that it does not bind to the *attL* and *attR* sites (Fig 3K). Deletion of the *oriT* site resulted in a significant decrease in excision frequency and a significant increase in extrachromosomal copy number, suggesting a potential role for AbiEi in maintaining the genetic stability of ICE*Ssu*HN105 by binding to the *oriT* site to prevent recognition by ICE-encoding proteins (Fig 3L and 3M).

## Identifying SezA as a regulator blocking the *attL* site to maintain the genetic stability of ICE*Ssu*HN105

Further analysis revealed a pair of palindromic sequences, named IR1 and IR2, within the SezAT promoter region (Fig 4A). EMSA analysis confirmed that the antitoxin SezA bound to the promoter and negatively regulated the transcription of SezAT (Fig 4B and 4D). Deletion of IR1 and IR2 prevented SezA from binding to the promoter and exerting negative regulatory effects (Fig 4C and 4D). The results indicated that SezA binds to the IR1 and IR2. To explore SezA's role in maintaining the genetic stability of ICE, we analyzed the transcription levels of genes in Δ*SezAT*. The data indicated that the deletion of SezAT did not affect transcription, indicating that SezA does not regulate the transcription of the target gene to maintain the genetic stability of ICE (S4 Fig). Following the analysis of T/L/R sequences, it was determined that the *attL* site and surrounding DNA sequences exhibited significant similarity to IR1 and IR2 of the SezAT promoter (Fig 4E). The EMSA analysis demonstrated that SezA exhibited dose-dependent binding to the L sequence (Fig 4F). The binding affinity of SezA for the L sequence was further analyzed by determining a $K_D$ of 1955.59 (Fig 4G). After the deletion of *attL*, SezA was unable to bind to the sequence, implying that SezA binds to the *attL* site (Fig 4H). Meanwhile, ChIP analysis demonstrated direct *in vivo* binding of SezA to the L sequence, with no observed binding to the T and R sequences (Fig 4I). The highly significant decrease in excision frequency after *attL* deletion suggests that SezA prevents ICE*Ssu*HN105 excision from the chromosome by binding to the *attL* site, which maintains the genetic stability of ICE (Fig 4J). Simultaneously, the extrachromosomal copy number of ICE decreased significantly to nearly zero following the blocked of excision (Fig 4K). To further explore the involvement of *OriT* and *attL* in ICE stability, a double deletion strain was constructed. Analysis of excision frequency and extrachromosomal copy number of ICE in double deletion strain revealed an inability for ICE excision and a reduction in extrachromosomal copy number to near zero levels (S6 Fig). These findings indicate the essential role of *OriT* and *attL* in maintaining the stability of ICE*Ssu*HN05.

## DNA-binding domains of SezA and AbiEi are required for maintaining the genetic stability of ICE*Ssu*HN105

To determine the role of antitoxin DNA-binding in ICE stability, two DNA-binding domain deletion mutants, Δ*SHTH* and Δ*AHTH*, were further constructed (Fig 5A and 5B). Analysis of the excision frequency and extrachromosomal copy number of ICE*Ssu*HN105 revealed a significant increase in excision frequency (Fig 5C) and a decrease in extrachromosomal copy number in the Δ*SHTH* and Δ*AHTH* mutants (Fig 5D). The results were consistent with antitoxin deletions and deletions of the TA system, suggesting that the DNA-binding domain of

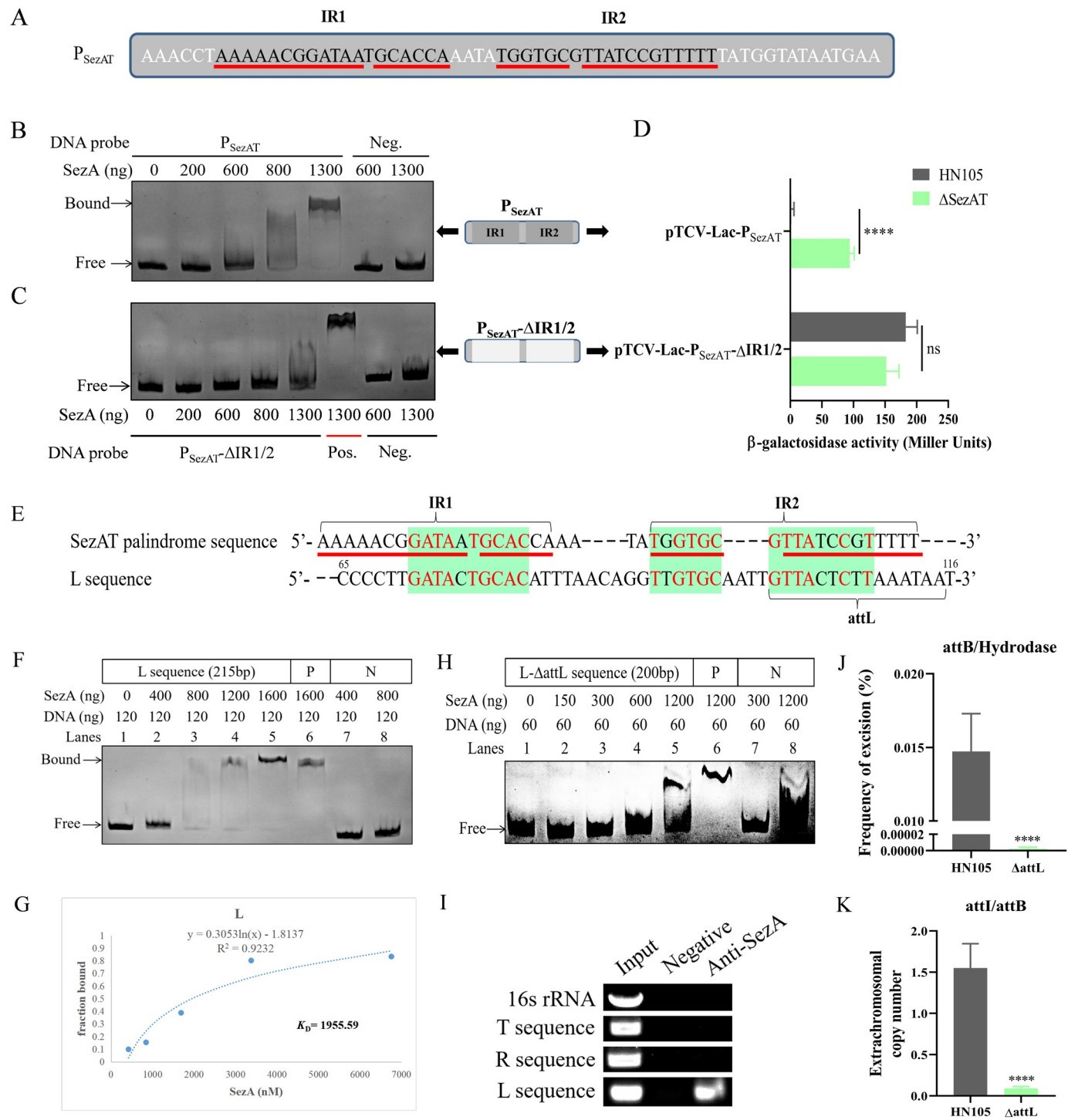

**Fig 4. Auto-regulatory factor SezA maintains the genetic stability of ICE*su*HN105 by binding to sequences in the *attL* site.** (A) Schematic representation of the SezAT promoter palindromic sequences IR1 and IR2. The red underlines indicate the position of the palindromic sequences. EMSA results shown the binding of SezA to the SezAT promoter (B) and no binding to P_AbiE-ΔIR1/2 (C). Different concentrations of SezA protein were added to each reaction mixture. DNA probes containing the fragments were used at 120 ng per reaction mixture. Fragments amplified from 16S rRNA were used as negative control. The SezAT promoter fragment served as positive control. (D) pTCV-*lac*-P_SezAT and pTCV-*lac*-P_SezAT-ΔIR1/2 were integrated to HN105 and Δ*SezAT* host to determine promoter activity, respectively. (E) Sequence similarity analysis of SezAT promoter IR1 and IR2 with the L sequence at *attL*. The red underlines indicate the position of the palindromic sequences. (F) EMSA result shown that SezA binds to the L sequence. The concentrations of SezA were 0 (0 ng), 675.68 (400 ng), 1351.4 (800 ng), 2027 (1200 ng), 2702.7 (1600 ng), 2702.7 (1600 ng), 675.68 (400 ng) and 1351.4 nM (800 ng). The SezAT promoter fragment served as positive control. (G) Binding affinity analysis of SezA protein for L sequence. The SezA concentrations used for EMSA were 0 (0 ng), 422.3 (250 ng), 844.59 (500 ng), 1689.2

(1000 ng), 3378.4 (2000 ng) and 6756.8 nM (4000 ng). (H) The EMSA result shown that SezA does not bind to the L-ΔattL sequence. The concentrations of SezA were 0 (0 ng), 253.38 (150 ng), 506.76 (300 ng), 1013.5 (600 ng), 2027 (1200 ng), 2027 (1200 ng), 506.76 (300 ng) and 2027 nM (1200 ng). The L sequence served as positive control. (I) ChIP analysis was performed to detect the binding of SezA to T, R, and L sequences. The excision frequency (J) and extrachromosomal copy number (K) of ICESsuHN105 in ΔattL were determined by qPCR. All experiments were conducted independently three times. Unpaired two-tailed Student's t-test: ns P > 0.05; **** P < 0.0001.

the antitoxin plays an important role in the stabilization of ICE. Furthermore, SezA^Del-SHTH and AbiEi^Del-AHTH proteins, lacking the DNA-binding domain, were expressed and purified. EMSA analysis confirmed that after the loss of the DNA-binding domain of the antitoxin, SezA^Del-SHTH was unable to bind to its own promoter and L sequence (Fig 5E and 5F), and AbiEi^Del-AHTH was also unable to bind to its own promoter and T sequence (Fig 5G and 5H). These results suggest that the DNA-binding domain of the antitoxin is essential for self-regulation and maintaining the stability of the ICE.

## The cross-link analysis between AbiEi and SezAT

Interactions and trans-activations are present among various TA systems [29]. Given the observed coordinated effect between SezAT and AbiE, we subsequently investigated the possibility of a cross-modulation between these two systems. We discovered that the IR1 sequence of the AbiE promoter exhibits high similarity to the IR2 sequence of the SezAT promoter (S7A Fig). SezAT did not exhibit regulatory activity on AbiE, and SezA demonstrated an inability to interact with the AbiE promoter, as evidenced by the assessment of β-galactosidase activity and EMSA analysis (S7B and S7C Fig). However, AbiE negatively regulates SezAT, and AbiEi is bound to the promoter of SezAT (S7D and S7E Fig). Furthermore, ChIP analysis further confirmed that AbiEi could also directly bind to the promoter of SezAT in vivo (S7F Fig). The transcript level of SezAT was significantly upregulated after antitoxin AbiEi deletion (S7G Fig). Collectively, the type IV TA system AbiE negatively regulates type II TA system SezAT transcription through direct binding to the SezAT promoter by the antitoxin AbiEi. This cross-regulation enables bacteria to optimize the coordinate effect of SezAT and AbiE, thereby maintaining ICE stability under antibiotic pressure and facilitating the acquisition of drug resistance.

## Diverse ICEs harbour SezAT and AbiE together with multidrug resistance genes against antibiotics pressures

A blast search of ICEs with complete sequences on the online database ICEberg identified a total of 352 TA systems encoded in 784 ICEs, including AbiE (252), SymER (153), and SezAT (31) (Fig 6A). The TA system exhibited the highest coding rate in the ICESa2603 family (93.75%), followed by the SXT/R391 (89.06%) and Tn5253 (88.24%) families (Fig 6B). Subsequent analysis of the coding rates of AbiE and SezAT in these three families, as well as in ICE which mediates drug resistance, revealed that AbiE is prevalent in all three families, whereas SezAT is exclusively present in the ICESa2603 and Tn5253 families (Fig 6C and 6D). Additionally, genetic evolutionary analyses revealed that the AbiEi antitoxin encoded by ICE was categorized into 11 branches, with the predominant ICEs being the unknown ICE family in branch XI, the SXT/R391 family in branch III, the Tn5253 family in branch IV, and the ICESa2603 family in branch V (Fig 6E). SezA was classified into three groups: the Tn5253 family in Group I, the unknown ICE family in Group II, and the ICESa2603 family in Group III (Fig 6F). These results suggest that the effects of AbiE and SezAT in promoting the stable inheritance of ICE may be broadly applicable in the Tn5253 and ICESa2603 families. To confirm this hypothesis, we conducted an analysis of ICESsuAH681, which contains genes

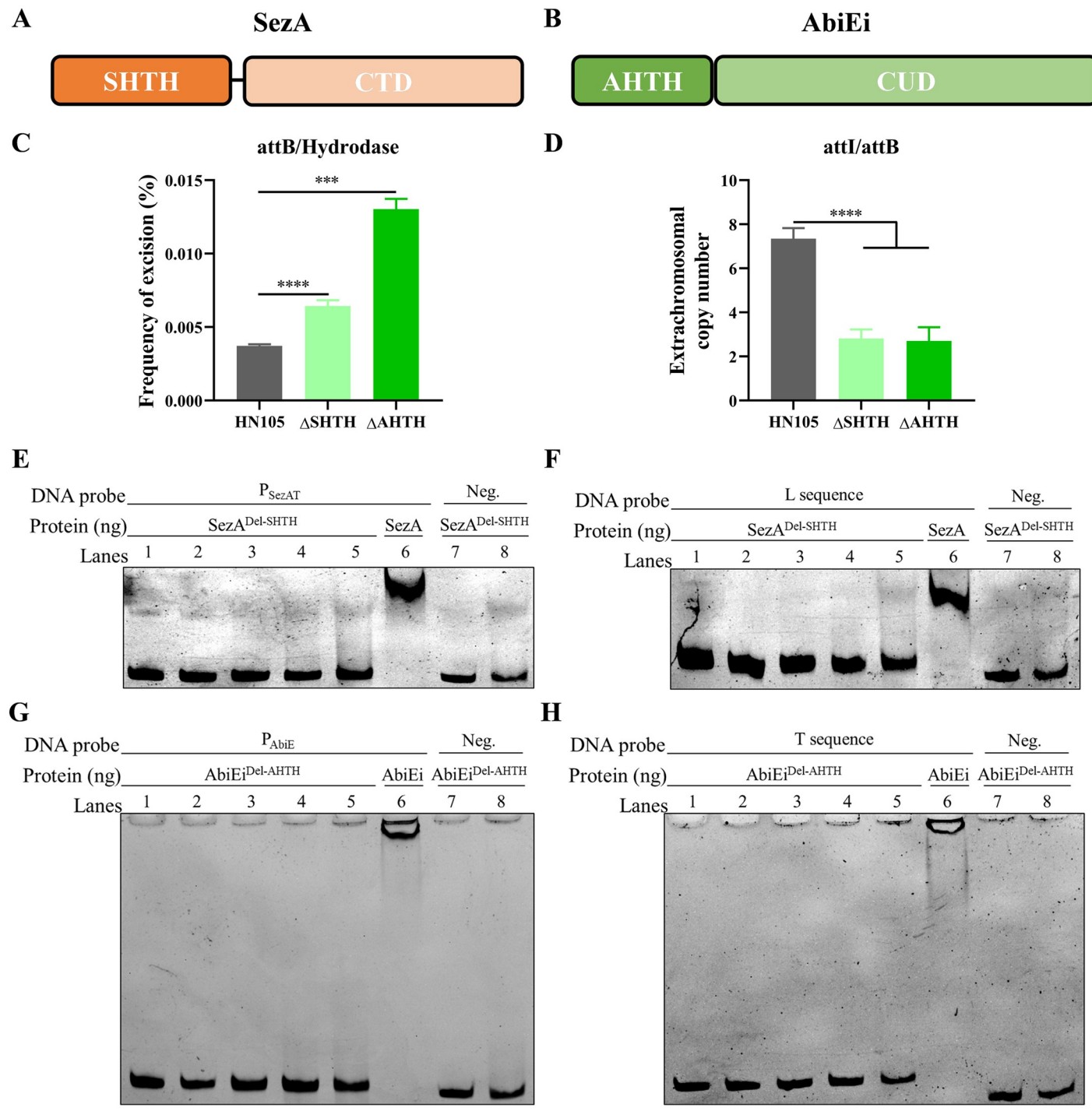

**Fig 5. The DNA-binding domains of the antitoxins SezA and AbiEi are required for maintaining the stability of ICE.** Schematic representation of the structural domains of SezA (A) and AbiEi (B). SHTH: SezA DNA-binding domain. CTD: C-terminal toxin neutralisation domain. AHTH: AbiEi DNA-binding domain. CUD: C-terminal unknown domain. The excision frequency (C) and extrachromosomal copy number (D) of ICE*Ssu*HN105 in Δ*SHTH* and Δ*AHTH* were determined by qPCR. The EMSA results shown that SezA[Del-SHTH] does not bind to the promoter (E) and L sequence (F). The concentrations of protein were 0, 400, 800, 1600, 3200, 3200, 800 and 3200 ng (Lanes 1–8). The SezA protein served as positive control. The EMSA results shown that AbiEi[Del-AHTH] does not bind to the promoter (G) and T sequence (H). The concentrations of protein were 0, 300, 600, 1200, 2400, 2400, 600 and 2400 ng (Lanes 1–8). The AbiEi protein served as positive control. DNA probes containing the fragments were used at 60 ng per reaction mixture. Fragments amplified from 16S rRNA were used as negative control.

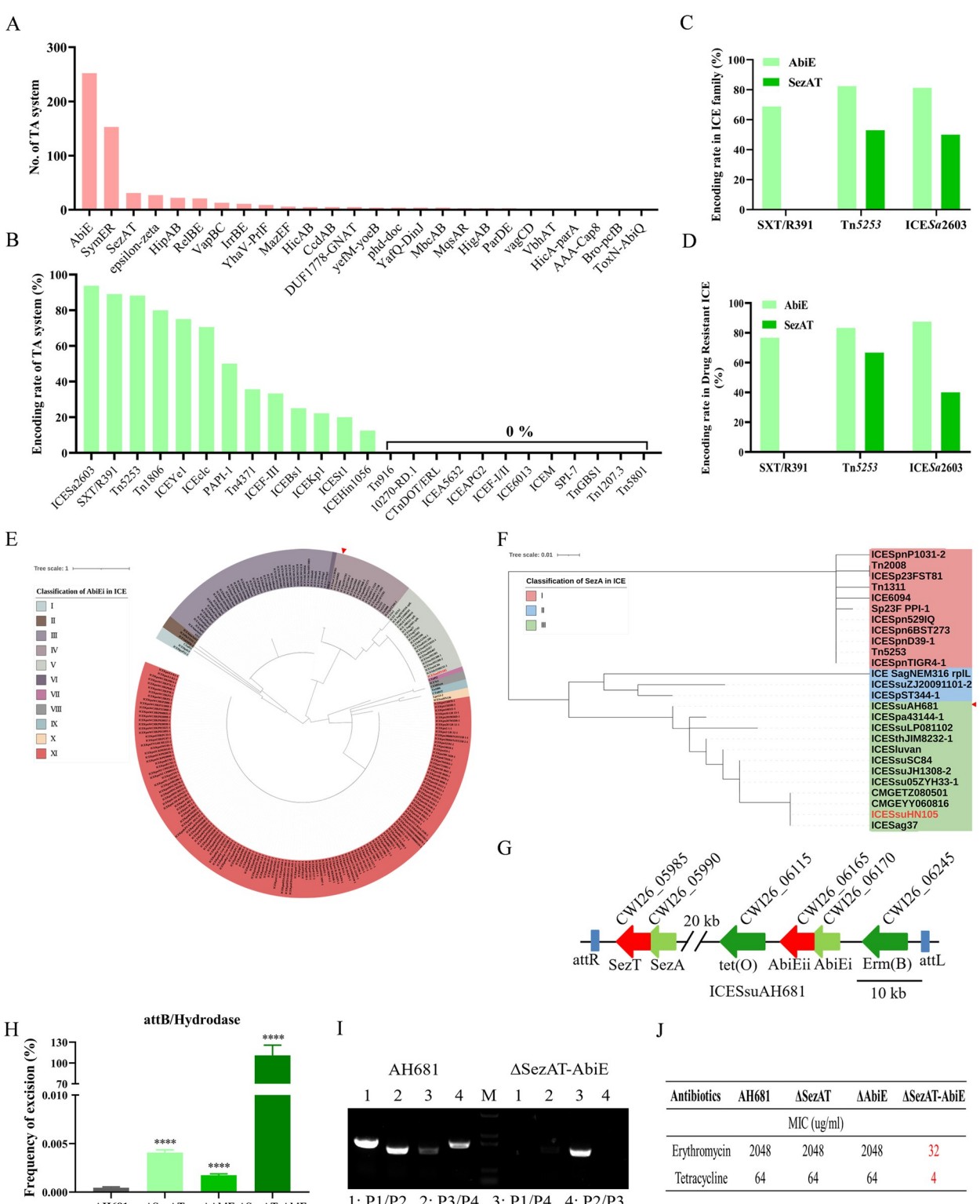

**Fig 6. The effect of SezAT and AbiE to synergistically maintain the genetic stability of ICE is also present in other mediating multidrug resistance ICEs.** (A) The blastp search of the TA system was performed in ICE with complete genomes on the ICEberg website. (B) Analysis of TA system coding rates in different ICE families. (C) Coding rates of SezAT and AbiE in the SXT/R391, ICE*Sa*2603 and Tn*5253* families. (D) Coding rates of SezAT and AbiE in mediating drug resistance ICE in the SXT/R391, ICE*Sa*2603 and Tn*5253* families. (E) The phylogenetic tree analysis of AbiEi in ICE. (F) The phylogenetic tree analysis of SezA in ICE. ICE*Ssu*HN105 is in red. Red triangle indicates ICE*Ssu*AH681. (G) Schematic

representation of the ICE*Ssu*AH681. Two TA systems, SezAT and AbiE, and two resistance genes, Erm(B) and tet(O) are encoded in ICE*Ssu*AH681. (H) The excision frequencies of ICE*Ssu*AH681 in AH681, AH681-Δ*SezAT*, AH681-Δ*AbiE* and AH681-Δ*SezAT-AbiE* were determined by qPCR. (I) PCR analysis of excision and circular forms of ICE*Ssu*AH681 in AH681 and AH681-Δ*SezAT-AbiE*. All experiments were conducted independently three times. Unpaired two-tailed Student's t-test: **** P < 0.0001. (J) Analysis of MIC values of erythromycin and tetracycline against AH681-Δ*SezAT-AbiE*.

encoding SezAT and AbiE, along with tet(O) and Erm(B) that confer drug resistance (Fig 6G). Subsequently, single and double deletion strains of SezAT and AbiE were constructed, and primers were designed for two standard curves to quantify the copy number of AH681-attB and AH681-hyd using qPCR (S8 Fig). By analyzing the excision frequency of ICE*Ssu*AH681 in the deletion strains, we found that both single deletions of SezAT and AbiE resulted in a significant increase in excision frequency (Fig 6H). Double deletion of SezAT and AbiE resulted in an excision frequency of ICE*Ssu*AH681 exceeding 100%, indicating that ICE*Ssu*AH681 was completely excised after double deletion of SezAT and AbiE (Fig 6H). PCR analysis showed that only ICE excision on the chromosome was detected in the double deletion strain, and no integration and circular forms were detected, indicating the loss of ICE*Ssu*AH681 (Fig 6I). The MIC values indicated that the single deletion of SezAT and AbiE did not affect the strains' antibiotic susceptibility, while the double deletion of SezAT and AbiE increased susceptibility (Fig 6J). These findings suggest that the SezAT and AbiE encoded by ICE*Ssu*AH681 work coordinately to maintain the genetic stability of ICE and mediate the acquisition of multidrug resistance, hinting at a similar coordinate effect of SezAT and AbiE in other ICEs.

## Discussion

ICEs are widely transferred between bacteria as one of the most abundant mobile genetic elements. These elements contain a diverse array of novel trait determinants, such as drug resistance genes, virulence factors, heavy metal resistance genes, and more [10]. ICE helps bacteria acquire key traits to resist environmental stress, and the transfer of ICE between bacterial populations facilitates evolutionary processes. Nevertheless, the precise mechanism underlying the stable inheritance of ICE within bacterial populations remains inadequately elucidated. In this study, the type II TA system SezAT and the type IV TA system AbiE, broadly encoded in ICE, are identified as coordinately promoting the stable inheritance of ICE and mediating the acquisition of multidrug resistance in *S. suis* (Fig 7).

TA systems, characterized as small genetic elements, are prevalent in bacterial and archaeal genomes. Bacterial genomes typically contain multiple TA systems, as exemplified by the genome of *M. tuberculosis*, which harbors over 80 TA systems [30]. In addition to performing their specific functions, there are interactions between homologous or non-homologous TA systems within the genome. Interactions between type II TA systems have been extensively characterized at both the level of toxicity neutralization and protein expression [29]. The toxins and antitoxins of the three homologous RelBE modules in *M. tuberculosis* can interact with each other and maybe cross-regulated [31]. Interactions between the toxin MazF and the antitoxin VapB have been identified in the non-homologous TA system of *M. tuberculosis* [32]. These interactions emphasize the intricate functional network of TA systems, but their exact role in bacterial life activities needs to be further explored. There is also trans-activation between non-homologous TA systems, triggering the expression of downstream TA systems through the degradation of mRNAs, mainly by non-homologous toxins that exert endoribonuclease activity [29]. For instance, the upregulation of the type II toxin Doc in *Escherichia coli* induces the activation of RelBE [33], while VapC triggers the activation of yefM-yoeB in *Shigella* [34]. The precise function of antitoxins in their interactions with either homologous or

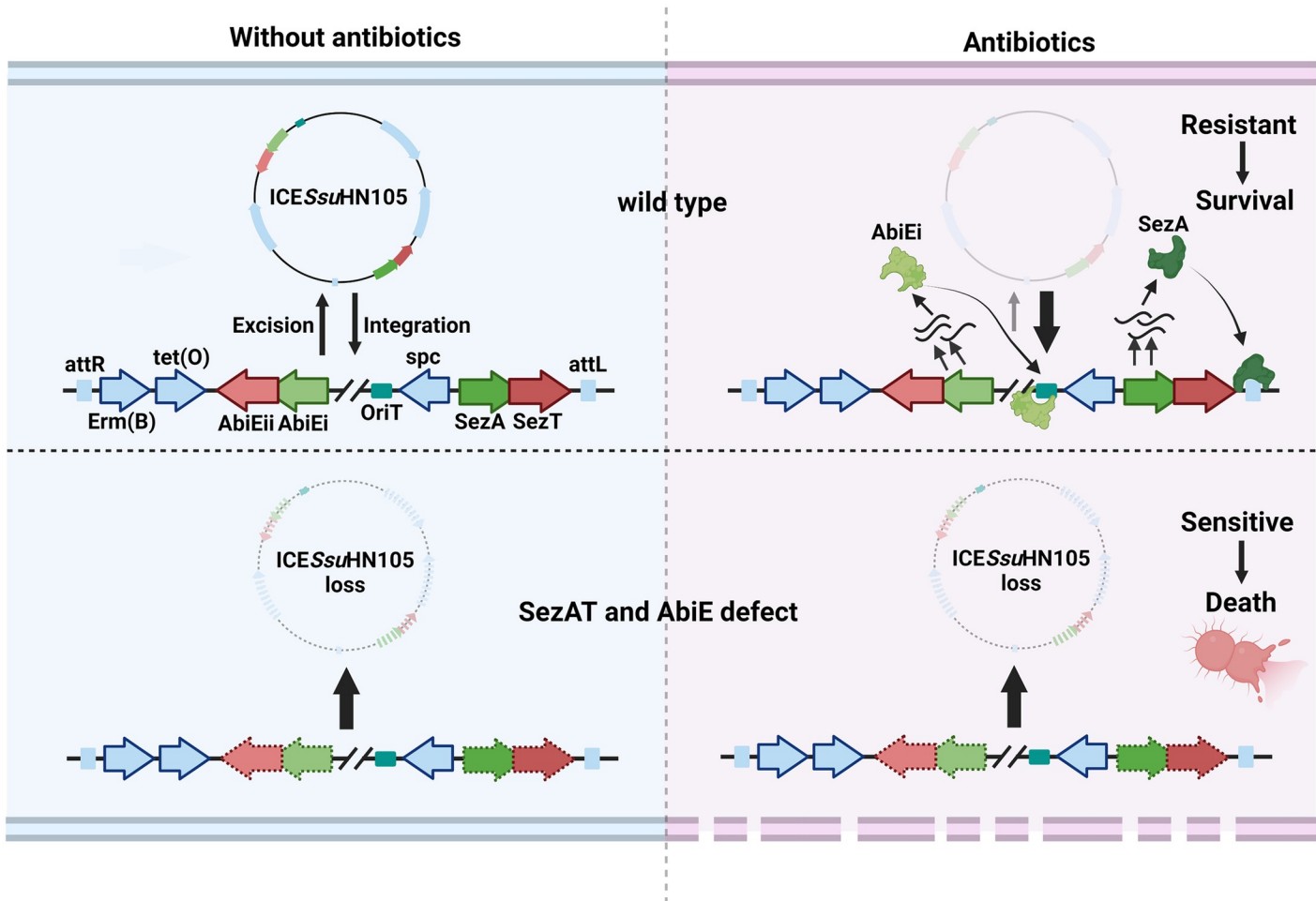

**Fig 7. Pattern diagram of SezAT and AbiE synergistically maintaining the genetic stability of ICE*Ssu*HN105.** Under normal conditions, the excision and integration of ICE*Ssu*HN105 are in dynamic equilibrium. When under antibiotics (Cargo gene-mediated resistance) pressures, SezAT and AbiE are activated, and highly expressed SezA and AbiEi bind to sequences in the *oriT* and *attL* sites, respectively, to inhibit excision and coordinately maintain the genetic stability of ICE*Ssu*HN105, which in turn helps *S. suis* to acquire antibiotic resistance. Inactivation of both SezAT and AbiE results in complete excision and loss of ICE*Ssu*HN105, making the bacteria susceptible to antibiotics. At this point, SezAT and AbiE double deficient strain is then killed after exposure to antibiotics.

non-homologous TA systems has yet to be fully elucidated. TA systems have been classified into eight types, and further study is needed to determine whether there are interactions between different types of TA systems. Our study elucidates the regulatory role of the antitoxin AbiEi from the type IV TA system in repressing the transcription of the type II TA system SezAT through direct binding to the SezAT promoter. Although the type II and type IV TA systems in *Klebsiella pneumoniae* exhibit distinct evolutionary patterns [35], our findings indicate a synergistic effect between the type II TA system SezAT and the type IV TA system AbiE. Our study complements the interactions of TA systems between different types and emphasizes the complex regulatory network of TA systems in bacterial physiological activities.

Numerous studies have shown that antitoxins can effectively bind to DNA targets to achieve negative regulatory effects, such as the type II antitoxin MqsA in *E. coli* [36], the antitoxin PezA in *Streptococcus pneumoniae* [37], and the type IV antitoxin AbiEi in *S. agalactiae* [21] and *M. tuberculosis* [38]. Although it has been shown that in *Staphylococcus aureus*, the formation of a complex between the toxin SavS and the antitoxin SavR enhances antitoxin binding

to the promoter [3], it has also been shown that the formation of a complex between the toxin HigB and the antitoxin HigA in *Pseudomonas aeruginosa* prevents the antitoxin DNA-binding process [4], suggesting that type II toxins may or may not act as co-repressors. In this study, further exploration is needed on the effect of the type II toxin SezT on DNA-binding capacity after forming a complex with the antitoxin SezA. On the other hand, the type IV toxin AbiEii was unable to interact with the antitoxin AbiEi. Consequently, the toxin AbiEii could not directly influence the DNA-binding domain of the antitoxin and affect its binding capacity.

Originally discovered on plasmids, the TA system was thought to be responsible for the stability of the plasmid [1]. Plasmid stabilization is maintained through the action of PSK, which functions to retain plasmids within the cell by killing dividing cells that have lost plasmids [12]. However, many studies argue this mechanism, with Tim F. Cooper et al. concluding that PSK does not contribute to plasmid stability and that the stabilization of plasmids encoding the TA system is achieved through plasmid-plasmid competition [39]. Plasmids serve as essential vectors in bacterial evolution, with the maintenance of plasmid stability through the TA system warranting further comprehensive investigation. Additionally, TA systems are present in diverse mobile genetic elements, including genomic islands and ICEs. The stabilization of genomic islands is facilitated by the TA system sgiAT encoded on the *Salmonella* genomic island [40] and SezAT encoded within the *S. suis* genome, promoting the stability of the SsPI-1 virulence island [20]. The TA system also contributes to the stability of the ICE SXT [14]. However, the precise mechanism by which the TA system promotes the maintenance of mobile genetic elements remains unclear. In this study, we elucidated how the type II TA and type IV TA systems, widely encoded in ICE, coordinately promote the genetic stability of ICE by investigating SezAT and AbiE encoded by *S. suis* ICE*Ssu*HN105. A comprehensive understanding of the mechanisms would help establish potential strategies to inhibit the evolution of bacterial drug resistance.

ICEs possess two defining features: integration into the host genome and encoding a type IV secretion system that mediates the transfer of ICEs between bacteria [10]. The integration of ICEs into the genome requires integrase, which inserts ICEs into specific attachment sites (*att*) on the bacterial chromosome. Integrase plays a crucial role in both the integration and excision of ICE. In addition, the ICE-encoded recombination directionality factor is also required for excision. In the SXT, the expression of the integrase is controlled by SetC and SetD [41]. Our findings indicate that the antitoxins SezA and AbiEi are also responsible for the regulation of ICE*Ssu*HN105 excision, as their deletion significantly increased the excision frequency. ICEs are attracting attention due to their widespread transfer via conjugation machinery mediated by the type IV secretion system. The process of conjugation necessitates the involvement of both the type IV secretion system and ICE-encoded DNA processing proteins, specifically relaxase, which is responsible for recognizing and binding to the *oriT* site [10,42]. Furthermore, it has been shown that the helicase processivity factor is also required for the conjugation of ICE*Bs1* [43]. The integration of ICEs into the genome typically results in the repression of excision and conjugation-related genes, which are subsequently activated in response to specific conditions, including the SOS response, the growth state of the host, and the selective advantage conferred by cargo genes [44–46]. Our findings indicated that the SezAT and AbiE systems are activated under conditions where ICE-carried resistance genes provide a selective advantage. This activation contributes to the stabilization of ICE and mediates the acquisition of multidrug resistance advantage. Although there has been extensive research on the integration and conjugation machinery of ICE, it is not well understood how ICE maintains stability. The maintenance of ICE stability throughout the life cycle is essential for bacterial populations to acquire a selective advantage and for bacterial evolution. This study elucidates the molecular mechanism through which the TA system maintains the genetic

stability of ICE and mediates the acquisition of multidrug resistance, providing a theoretical basis for the development of strategies to combat the spread of drug resistance.

In summary, our study demonstrates that the TA system maintains the genetic stability of ICE and mediates the acquisition of multidrug resistance in *S. suis*. This mechanism is coordinately mediated by the type II TA system SezAT and the type IV TA system AbiE, which are widely encoded in the ICE*Sa*2603 family and the Tn*5253* family ICE. SezAT and AbiE facilitate ICE stabilization through their antitoxins playing a regulator-like role in binding to several critical sites of ICE integration. This novel mechanism may explain the high rates of erythromycin, lincomycin, and tetracycline resistance among streptococci. The way in which bacteria acquire a selective advantage by using the TA system to maintain the genetic stability of ICE opens a novel perspective in our understanding of bacterial evolution. Additionally, our study also provides important targets for antimicrobial therapy.

## Materials and methods

### Bacterial strains, plasmids and growth conditions

The strains and plasmids utilized in this study are detailed in S1 Table in the supplementary material. *S. suis* was cultured in Todd Hewitt broth (THB, Oxoid Cheshire, United Kingdom) at 37˚C. Growth of *E. coli* was carried out at 37˚C on Luria-Bertani (LB) broth or LB agar plates. For mutant selection, if required, the THB or THA was supplemented with chloramphenicol (10 μg/ml). Antibiotics [chloramphenicol (10 μg/mL), kanamycin (50 μg/mL), or ampicillin (100 μg/mL)] were added to the LB medium to maintain plasmids when necessary.

### Antimicrobial susceptibility tests

The minimum inhibitory concentrations (MICs) of antibiotics against *S. suis* were determined following guidelines established by the Clinical and Laboratory Standards Institute [47]. *S. suis* was cultured to $OD_{600} = 0.6$ (log phase) and subsequently diluted 1000-fold with fresh THB. A volume of 180 μL of the inoculum culture was added to the first well, with 100 μL added to the remaining wells in a 96-well microtiter plate. Following the addition of 20 μL of antibiotics to the first well and thorough mixing, 100 μL of the mixture was transferred to the subsequent wells up to the 10th well. Wells 11 and 12 were positive and negative controls, respectively. After 20 hours of incubation at 20˚C, results were recorded.

### Construction of mutant strains

Mutants were created by the natural DNA transformation and a few modifications [48]. The primer sequences can be found in S2 Table of the supplemental material. The upstream and downstream sequences were amplified from the *S. suis* genome. Overlapping PCR was used to fuse the upstream and downstream fragments with sacB-CM cassettes. Synthetic peptides and fusion fragments were added to 100 μL of bacteria ($OD_{600} \approx 0.042$) and incubated for 2 h at 37˚C before being plated in THA containing chloramphenicol. The negative control, sacB, is susceptible to sucrose. The positive mutant was then constructed with the fusion homologous fragment without a cassette. THB plates containing 10% (w/v) sucrose were then used to maintain the transformed bacteria. Positive clones were identified through PCR analysis.

### Growth curve determination and CFU assay

Top10 was grown in LB containing 100 μg/ml ampicillin and 0.2% L-arabinose was added to the medium when $OD_{600} = 0.3 \sim 0.5$. BL21 (DE3) was cultured in LB medium with 50 μg/mL kanamycin and 10 μg/mL chloramphenicol. Additionally, 1 mM IPTG and 0.2% L-arabinose

were added when the $OD_{600}$ = 0.3 ~ 0.5. $OD_{600}$ was measured every two hours and the results were recorded. In the colony-forming unit (CFU) assay, *E. coli* was grown to an $OD_{600}$ = 0.6, diluted with sterile PBS, and plated on LB agar containing 0.2% L-arabinose or 0.2% L-arabinose and 1mM IPTG. The plates were placed in an incubator set at 37˚C for overnight incubation, followed by subsequent observation of the results.

## RNA isolation and qRT-PCR

Whole bacterial RNA was extracted from the log phase using TRIzol (Vazyme, China), followed by cDNA synthesis after removal of contaminating DNA using a gDNA wiper. Transcription levels of genes were determined using the QuantStudio 6 Flex RT-PCR system and ChamQ Universal SYBR qPCR Master Mix (Vazyme, China), with the housekeeping gene *parC* serving as an internal reference [49]. Relative fold changes were calculated using $2^{-\Delta\Delta CT}$ [50].

## Promoter activity assay

The pTCV-lac-$P_{AbiE}$, pTCV-lac-$P_{AbiE}$-ΔIR1, pTCV-lac-$P_{AbiE}$-ΔIR2, and pTCV-lac-$P_{AbiE}$-ΔIR1/2 plasmids were introduced into HN105 and Δ*abiE* strains. Additionally, the pTCV-lac-$P_{SezAT}$ and pTCV-lac-$P_{SezAT}$-ΔIR1/2 plasmids were transferred into HN105 and Δ*sezAT* strains. The measurement of β-galactosidase activity was conducted using Miller's method with slight modifications [51]. Following growth to the log phase, 2 mL of cell culture was centrifuged, washed twice with sterile PBS and the precipitate was resuspended in 200 μL of precooled z-buffer containing 2.5 mg/mL lysozyme and 50 mM β-mercaptoethanol. Then, chloroform and 0.1% SDS were added and mixed, and 100 μL o-nitrophenyl-β-d-galactoside (4 mg/mL) was added after 5 min in a 30˚C water bath, and the reaction was maintained in a 30˚C water bath until the solution was no longer yellow. The reaction was terminated by the addition of 250 μL of sodium carbonate, followed by the transfer of 250 μL of the supernatant to a 96-well plate. Absorbances at 420 nm and 550 nm were recorded using a microplate reader, and β-galactosidase activity was calculated using the following formula: Activity [MU] = $[1{,}000 \times (A\ 420 - 1.75 \times A\ 550)] / [t(min) \times 0.1 \times OD\ 600]$, where MU = Miller units; t = reaction time. At least three independent cultures of each strain were tested in each experiment.

## Antitoxin expression and purification and polyclonal antibody preparation

The coding sequences of SezA and AbiEi were amplified from the genome and ligated into the pET28a-SUMO plasmid to produce pET28a-SUMO-SezA and pET28a-SUMO-AbiEi. BL21 (DE3) was transformed with pET28a-SUMO-SezA or pET28a-SUMO-AbiEi plasmids and grown to $OD_{600}$ of 0.4–0.6. A 1 mM solution of isopropyl-β-D thiogalactopyranoside (IPTG) was added and the cells were cultured at 16˚C for 16 hours. The cells were harvested and sonicated in lysate buffer (20 mM Na3PO4-12H2O, 0.5 mM NaCl, 30 mM imidazole, pH 7.4), and SezA and AbiEi proteins were purified using a His-tagged Ni-NTA column. Proteins were eluted using imidazole with concentrations between 50 and 500 mM and protein concentrations were quantified using the BCA method. BALB/c mice were then immunized with 100 μg of purified SezA and AbiEi proteins via subcutaneous injection after emulsification with adjuvant, respectively. Mice serum was collected 10 days following the third immunization.

## EMSA

EMSA was conducted using native polyacrylamide gels and fragment probes. The negative-control probe was amplified from the coding sequence of the 16s. Various concentrations of proteins and DNA probes were incubated in binding buffer (10 mM Tris, 50 mM KCl, 5 mM MgCl2, 1 mM dithiothreitol, 0.05% Nonidet P-40, 2.5% glycerol, pH 7.5) at 37˚C for 42 min. Electrophoresis was carried out at 200 V for 45 mins using a 6% natural polyacrylamide gel in 0.5× TBE buffer (44.5 mM Tris-base, 44.5 mM boric acid, 1 mM EDTA, pH 7.5). The gels were then stained with SYBR Green I and photographed according to the instructions (SBS Gene-tech, China). To determine the binding affinity of the antitoxin for DNA, the equilibrium dissociation constants ($K_D$) were determined by EMSA according to the reference [52].

## ChIP assay

ChIP analyses were carried out according to reference with minor modifications [53]. Bacteria cells were fixed with formaldehyde at a final concentration of 1% for 10 min and then cross-linking was terminated with glycine at a final concentration of 0.125 M. Cells were resuspended in lysis buffer and then sonicated to fragment chromosomal DNA in the range of 0.5–1.0 kb. Following centrifugation, anti-SezA, anti-AbiEi antibody, or negative serum was added and incubated at 4˚C overnight, then protein A/G beads (Beyotime, Shanghai, China) were incorporated and incubated for 2 h. The immunoprecipitation complexes were collected, washed twice with wash buffer and TE buffer, and finally eluted with elution buffer. Protein and RNA were removed by adding proteinase K and Rnase (TAKARA, Dalian, China), and DNA was isolated using a DNA fragment pure kit (Vazyme, China). The isolated DNA was used for PCR analysis.

## Detection of excision and Circularization of ICESsuHN105 and ICESsuAH681

The integrated and extrachromosomal circular forms of ICE*Ssu*HN105 and ICE*Ssu*AH681 were detected by different combination primers, P1-P4.

## Determination of ICE dynamics using real-time quantitative PCR

Excision frequency and extrachromosomal copy number were assessed by real-time quantitative PCR following the methodology outlined in the literature [27]. Primers were designed to amplify the attI, attB, and hydrodase fragments, which were then cloned into the pCE2-TA/Blunt-zero plasmid to create three standard curves for quantitative analysis. For the analysis of excision frequency and extrachromosomal copy number by extraction of strain DNA and qPCR, excision frequency was calculated using the formula: attB/hydrodase × 100% and extrachromosomal copy number: attI/attB. All qPCR experiments were carried out in triplicate wells in three independent experiments.

## Bacterial two-hybrid assays

The Bacterial adenylate cyclase two-hybrid (BACTH) system kit from Euromedex was employed for conducting bacterial two-hybrid assays. Recombinant plasmids were obtained by amplifying the coding sequences of AbiEi and AbiEii and ligating them to pKT25 and pUT18C by restriction digestion, respectively. pKT25-AbiEi and pUT18C-AbiEii were co-transformed into the BTH101 strain, which serves as the reporter strain for the BACTH assay. X-Gal (5-bromo-4-chloro-3-indolyl-β-D-galactopyranoside, 40 μg/ml) and IPTG (0.5 mM) were added to the positive clone medium and detected at 30˚C for 24–72 hours. The plasmids

pKT25-zip and pUT18C-zip were used as positive controls, while pKT25 and pUT18C served as negative controls. The experimental results were then recorded.

## Statistical analyses

Each experiment was repeated at least three times. GraphPad Prism 8 was used for analysis and presentation. Data were analyzed using an unpaired two-tailed Student's t-test or a log-rank test (Mantel-Cox). Statistical significance was set at $P < 0.05$.

## Supporting information

**S1 Data. Excel spreadsheet containing, in separate sheets, the underlying numerical data for Figs 1C, 1E, 1F, 2C, 2D, 2G, 2H, 3F, 3J, 3L, 3M, 4D, 4G, 4J, 4K, 5C, 5D, 6A, 6B, 6C, 6D, 6H, 6J, S1B, S1F, S2B, S2C, S2D, S3A, S3B, S4, S6A, S6B, S7B, S7D, S7G, S8A, and S8B.** (XLSX)

**S2 Data. File containing, in zip folder, the original uncropped pictures for Figs 1B, 1D, 2B, 2E, 2F, 3B, 3C, 3D, 3E, 3H, 3I, 3J, 3K, 4B, 4C, 4F, 4G, 4H, 4I, 5E, 5F, 5G, 5H, 6I, S1A, S1E, S1G, S7C, S7E, and S7F.** Two text files containing all the amino acid sequences for Fig 6E and 6F. (ZIP)

**S1 Fig. Zeta-epsilon identification and analysis of the Type IV TA system AbiE.** (A) CFUs were analyzed for the toxicity of zeta and the ability of epsilon to neutralize toxicity. (B) Growth curves for cells transfected with pBAD33/pET28a, pBAD33-zeta/pET28a, pBAD33/pET28a-epsilon or pBAD33-zeta/pET28a-epsilon vectors were determined under L-arabinose and IPTG induction. (C) Tertiary structure alignment of the AbiEi antitoxin in HN105 with the AbiEi antitoxin (6y8q.1.A) in Streptococcus agalactiae. AbiEi in HN105 overlaps almost completely with the tertiary structure of 6y8q.1.A. The amino acid sequence identity between AbiEi in HN105 and 6y8q.1.A is 88.21%. (D) AbiEii in HN105 largely overlaps with the tertiary structure of the TglT toxin (guanylytransferase-like toxin, 6j7q.1.A) in *Mycobacterium tuberculosis*. The amino acid sequence identity between AbiEii and 6j7q.1.A is 15.73%. (E) CFUs were analyzed for the toxicity of AbiEii and the ability of AbiEi to neutralize toxicity. (F) CFUs assay for the identification of AbiE activity using pBAD33 and pET28a co-transformation. (G) The interaction between AbiEi and AbIEii was verified by bacterial two-hybrid analysis. 1–3 are three independent replicates. (TIF)

**S2 Fig. Establishment of standard curves.** (A) Schematic representation of the excision and integration of ICE*Ssu*HN05. Bold black short lines represent the locations of attI, attB and Hydrodase fragment amplification. Standard curves for attI (B), attB (C) and Hydrodase (D) copy number analysis. (TIF)

**S3 Fig. Transcript levels of toxins and antitoxins were analyzed in deletion strains.** (A) Transcript levels of SezA and SezT were determined in Δ*SezA* and Δ*SezT*. (B) Transcript levels of AbiEi and AbiEii were determined in Δ*AbiEi* and Δ*AbiEii*. All experiments were conducted independently three times. Unpaired two-tailed Student's *t*-test: *** $P < 0.001$; **** $P < 0.0001$. (TIF)

**S4 Fig. Analysis of transcript levels of ICE excision and integration-related genes in Δ*SezAT*, Δ*AbiE* and Δ*SezAT-AbiE*.** Deletion of SezAT and AbiE does not affect the transcription

of excision and integration-related genes.
(TIF)

**S5 Fig. T (A), L (B) and R (C) sequences on the genome.** Grey areas represent the sites for *oriT*, *attL*, and *attR*, respectively. Red arrows indicate the predicted recognition and cleavage sites of the relaxase.
(TIF)

**S6 Fig. The excision frequency (A) and extrachromosomal copy number (B) of ICES-suHN105 in ΔOriT-attL were determined by qPCR.** All experiments were conducted independently three times. Unpaired two-tailed Student's *t*-test: **** $P < 0.0001$.
(TIF)

**S7 Fig. The AbiEi antitoxin has a negative regulatory effect on SezAT transcription by directly binding to SezAT's promoter.** (A) Sequence similarity analysis of AbiE promoter IR1 and IR2 with the SezAT promoter IR1 and IR2. (B) pTCV-*lac*-P$_{AbiE}$ was integrated to HN105 and Δ*SezAT* host to determine promoter activity. (C) The EMSA result shown that SezA does not bind to the AbiE promoter. The SezAT promoter served as positive control. (D) pTCV-*lac*-P$_{SezAT}$ was integrated to HN105 and Δ*AbiE* host to determine promoter activity. (E) EMSA result shown that AbiEi binds to the SezAT promoter. The AbiE promoter fragment served as positive control. (F) ChIP analysis was performed to detect the binding of AbiEi to SezAT promoter. The Anti-SezA antibody served as a positive control. Normal mouse serum was used as a negative control. The 16S rRNA gene PCR product was used as a negative control. (G) Transcript levels of SezA and SezT in Δ*AbiEi* and Δ*AbiEii*. All experiments were conducted independently three times. Unpaired two-tailed Student's *t*-test: ns $P > 0.05$; *** $P < 0.001$; **** $P < 0.0001$.
(TIF)

**S8 Fig. Establishment of standard curves.** Standard curves for AH681-attB (A) and AH681-hyd (B) copy number analysis.
(TIF)

**S1 Table. Bacterial strains and plasmids used in this study.**
(DOCX)

**S2 Table. All primers used in this study.**
(DOCX)

## Acknowledgments

The authors are grateful to the OIE Reference Laboratory for Swine Streptococcosis, College of Veterinary Medicine, Nanjing Agricultural University for the provision of facilities for this study. We are also very appreciative of the help of all the staff.

## Author Contributions

**Conceptualization:** Qibing Gu, Xiayu Zhu, Zihao Pan, Jiale Ma, Huochun Yao.

**Data curation:** Qibing Gu, Xiayu Zhu, Jiale Ma.

**Formal analysis:** Qibing Gu, Zihao Pan, Jiale Ma, Huochun Yao.

**Funding acquisition:** Qibing Gu, Huochun Yao.

**Investigation:** Qibing Gu, Zihao Pan, Jiale Ma, Huochun Yao.

**Methodology:** Qibing Gu, Xiayu Zhu, Tao Jiang.

**Project administration:** Qibing Gu, Zihao Pan, Jiale Ma, Huochun Yao.

**Resources:** Qibing Gu, Yong Yu, Zihao Pan, Jiale Ma, Huochun Yao.

**Software:** Qibing Gu, Xiayu Zhu, Jiale Ma.

**Supervision:** Zihao Pan, Jiale Ma, Huochun Yao.

**Validation:** Qibing Gu, Yong Yu, Zihao Pan, Jiale Ma, Huochun Yao.

**Visualization:** Qibing Gu, Tao Jiang, Jiale Ma, Huochun Yao.

**Writing – original draft:** Qibing Gu, Jiale Ma, Huochun Yao.

**Writing – review & editing:** Qibing Gu, Zihao Pan, Jiale Ma, Huochun Yao.

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
