## [Decision Letter · Decision Letter 0]

26 Feb 2024

Dear Dr. yao,

Thank you very much for submitting your manuscript "Two types of toxin-antitoxin systems coordinately stabilize the integrative and conjugative element conferring multiple drug resistance" for consideration at PLOS Pathogens. As with all papers reviewed by the journal, your manuscript was reviewed by members of the editorial board and by independent reviewers. The reviewers appreciated the attention to an important topic. Based on the reviews, we are likely to accept this manuscript for publication, providing that you modify the manuscript according to the review recommendations.

Specifically, your manuscript was evaluated by three experts in the field. All three reviewers find your work to be interesting, well-presented, of high quality and rigor with broad implications for antibiotic resistance and bacterial evolution. The reviewers have mainly minor comments for you to address, with the exception of Reviewer #2, who provides comments and suggestions for additional experiments to solidify your claims regarding AbiEi and SezA-binding and ICE stability. I would ask that you, in some way, address and respond to the comments and suggestions from Reviewer #2 in a revised manuscript through incorporating additional data or discussing the issues raised in by the Reviewer in the text.

Sincerely,

Anders P Hakansson, Ph.D.

Academic Editor

PLOS Pathogens

Michael Wessels

Section Editor

PLOS Pathogens

Michael Malim

Editor-in-Chief

PLOS Pathogens

orcid.org/0000-0002-7699-2064

Your manuscript has been evaluated by three experts in the field. All three reviewers find your work to be interesting, well-presented, of high quality and rigor with broad implications for antibiotic resistance and bacterial evolution. The reviewers have mainly minor comments for you to address, with the exception of Reviewer #2, who provides comments and suggestions for additional experiments to solidify your claims regarding AbiEi and SezA-binding and ICE stability. I would ask that you, in some way, address and respond to the comments and suggestions from Reviewer #2 in a revised manuscript through incorporating additional data or discussing the issues raised in by the Reviewer in the text.

Reviewer Comments (if any, and for reference):

Reviewer's Responses to Questions

**Part I - Summary**

Reviewer #1: The manuscript is well presented in general. The amount of work is overwhelming, and very interesting results are convincing and well discussed. I have only a few comments that I would like the authors to consider.

Reviewer #2: This manuscript investigates how two antitoxins, SezA and AbiEi, promote the stability of an integrative and conjugative element (ICE) in the genome of Streptococcus suis. The antitoxins, rather than the toxins, were found to be important for maintaining the genetic stability of the ICE. AbiEi was found to interact with the oriT, and SezA was found to bind in the attL site. This interaction likely impedes the excision and subsequent loss of the ICE, highlighting a novel mechanism of ICE stabilization mediated by antitoxins. The results are exciting. However, there are a few issues that lessen the impact of the work.

Reviewer #3: Upon reviewing the manuscript focused on the study of toxin-antitoxin (TA) systems, particularly SezAT and AbiE, within Streptococcus suis, I can evaluate the depth of the research and its contributions to our understanding of bacterial resistance mechanisms. The detailed examination of the cross-regulation between two distinct TA pairs and their influence on the stability of integrative and conjugative elements (ICEs) is noteworthy. This aspect of the study not only highlights a novel and critical mechanism but also sheds light on the broader implications for antibiotic resistance and bacterial evolution.

The authors have provided robust evidence to support their findings, demonstrating a high level of scientific rigor and innovation. Their work addresses a gap in the current literature, offering new insights that could pave the way for developing more effective strategies against antibiotic-resistant bacteria.

Given the manuscript's clear relevance to the ongoing challenges in microbiology and public health, its potential to inform future research, and its contribution to the scientific community's understanding of complex bacterial systems, I recommend its publication in PLOS pathogens. The study is well-presented, methodologically sound, and represents an advancement in the field of microbial genetics and antibiotic resistance.

In conclusion, this manuscript is not only timely but also of importance to researchers, and clinicians involved in combating antibiotic resistance. Its publication would enrich the academic discourse and inspire further research in this study area.

**Part II – Major Issues: Key Experiments Required for Acceptance**

Reviewer #1: None

Reviewer #2: The authors propose an intriguing interaction between AbiEi-oriT and SezA-AttL contributing to the stabilization of ICE elements. However, the evidence presented does not fully substantiate this hypothesis. To bolster their claims, the authors should consider generating deletion mutants for the DNA-binding domains of AbiEi and SezA. This would provide crucial insights into the role of antitoxin DNA-binding in ICE stability. Furthermore, a double deletion mutant involving oriT and attL, in the genetic background of ∆SezA-AbiEi mutants, would be invaluable in supporting the hypothesis if the ICE stability could be restored.

What is the extrachromosomal copy number of the attL deletion mutant?

Reviewer #3: (No Response)

**Part III – Minor Issues: Editorial and Data Presentation Modifications**

Reviewer #1: GENERAL:

1. I would recommend being more specific in the title: type of TA, bacterial species, and type of ICE. As it is now, it has little information.

2. Although in general the English is correct, it needs editing throughout.

SPECIFIC:

3. Lines 140 and around it: a recent review on the hypothesis and roles of type II TAs, with a focus focusing on PezAT should be be cited (https://doi.org/10.1093/femsre/fuad052).

4. Line 172: “Numerous studies…” but they cite just two papers. Correct this or cite at least 3-5 more papers.

5. Line 251 and following: In many cases, the antitoxin binds poorly to its DNA target, needing the toxin as a co-repressor, which does not seem to be the case here. It would be interesting if the authors discuss this.

6. Discussion: lines 440-450: Cite the work on ICESag-rpsl, which encodes a functional relaxase involved in the mobilization of the element and also in plasmid mobilization (https://doi.org/10.1098/rsob.160084)

Reviewer #2: In Figures 3H and 3I, the omission of protein concentrations leaves a gap in understanding the experimental setup, especially in estimating the dissociation constant. The inclusion of these details would significantly enhance the interpretability of the results.

The Kd values mentioned on lines 288 and 315 warrant a second look, as transcription factors are known to bind DNA with nanomolar affinity. This discrepancy suggests a possible miscalculation or misinterpretation that should be addressed to ensure the accuracy of the findings.

Line 82, please provide the full name of SXT.

All of the figures' resolutions are too low.

Reviewer #3: (No Response)

PLOS authors have the option to publish the peer review history of their article (what does this mean?). If published, this will include your full peer review and any attached files.

Reviewer #1: **Yes: **Manuel Espinosa

Reviewer #2: No

Reviewer #3: No

Figure Files:

Data Requirements:

Reproducibility:

References:

---

## [Editor Report · Decision Letter 1]

2 Apr 2024

Dear Dr. yao,

We are pleased to inform you that your manuscript 'Type Ⅱ and Ⅳ toxin-antitoxin systems coordinately stabilize the integrative and conjugative element of the ICESa2603 family conferring multiple drug resistance in Streptococcus suis' has been provisionally accepted for publication in PLOS Pathogens.

Best regards,

Anders P Hakansson, Ph.D.

Academic Editor

PLOS Pathogens

Michael Wessels

Section Editor

PLOS Pathogens

Michael Malim

Editor-in-Chief

PLOS Pathogens

orcid.org/0000-0002-7699-2064
---

## [Editor Report · Acceptance letter]

16 Apr 2024

Dear Dr. Yao,

We are delighted to inform you that your manuscript, "Type Ⅱ and Ⅳ toxin-antitoxin systems coordinately stabilize the integrative and conjugative element of the ICESa2603 family conferring multiple drug resistance in Streptococcus suis," has been formally accepted for publication in PLOS Pathogens.

Best regards,

Michael Malim

Editor-in-Chief

PLOS Pathogens

orcid.org/0000-0002-7699-2064